# Mitochondrial Genome of *Fagopyrum esculentum* and the Genetic Diversity of Extranuclear Genomes in Buckwheat

**DOI:** 10.3390/plants9050618

**Published:** 2020-05-12

**Authors:** Maria D. Logacheva, Mikhail I. Schelkunov, Aleksey N. Fesenko, Artem S. Kasianov, Aleksey A. Penin

**Affiliations:** 1Institute for Information Transmission Problems of the Russian Academy of Sciences, 127051 Moscow, Russia; m.shchelkunov@skoltech.ru (M.I.S.); artem.kasianov@gmail.com (A.S.K.); alekseypenin@gmail.com (A.A.P.); 2Skolkovo Institute of Science and Technology, 143026 Moscow, Russia; 3Federal Scientific Center of Legumes and Groat Crops, 302502 Orel, Russia; fesenko.a.n@rambler.ru

**Keywords:** mitochondrial genome, buckwheat, plastid genome, genetic diversity, long reads, targeted assembly

## Abstract

*Fagopyrum esculentum* (common buckwheat) is an important agricultural non-cereal grain plant. Despite extensive genetic studies, the information on its mitochondrial genome is still lacking. Using long reads generated by single-molecule real-time technology coupled with circular consensus sequencing (CCS) protocol, we assembled the buckwheat mitochondrial genome and detected that its prevalent form consists of 10 circular chromosomes with a total length of 404 Kb. In order to confirm the presence of a multipartite structure, we developed a new targeted assembly tool capable of processing long reads. The mitogenome contains all genes typical for plant mitochondrial genomes and long inserts of plastid origin (~6.4% of the total mitogenome length). Using this new information, we characterized the genetic diversity of mitochondrial and plastid genomes in 11 buckwheat cultivars compared with the ancestral subspecies, *F. esculentum* ssp. *ancestrale*. We found it to be surprisingly low within cultivars: Only three to six variations in the mitogenome and one to two in the plastid genome. In contrast, the divergence with *F. esculentum* ssp. *ancestrale* is much higher: 220 positions differ in the mitochondrial genome and 159 in the plastid genome. The SNPs in the plastid genome are enriched in non-synonymous substitutions, in particular in the genes involved in photosynthesis: *psbA*, *psbC*, and *psbH*. This presumably reflects the selection for the increased photosynthesis efficiency as a part of the buckwheat breeding program.

## 1. Introduction

In contrast to plastid genomes, plant mitochondrial genomes are usually large and complex. They are shaped by multiple structural changes, including rearrangements, duplications, and horizontal gene transfer (HGT) from nuclear and plastid genomes and, in rare cases, from other plant species [1,2]. This complexity leads to the under-representation of the plant mitochondrial genome sequences compared to the plastome sequences. Even in the species for which complete nuclear genomes are characterized, the information on the mitochondrial genome is often lacking or incomplete. The subject of our study, *Fagopyrum esculentum*, is an important agricultural non-cereal grain plant. Three nuclear genome assemblies of *Fagopyrum* species are available by now ([3,4], and an unpublished study, accession number GCA_004303065), as well the number of plastid genomes [5,6]. However, not a single mitogenome sequence is available. At a larger scale, *Fagopyrum* belongs to the family Polygonaceae, an isolated group within the order Caryophyllales. By now the only species of Polygonaceae with a sequenced mitochondrial genome is *Fallopia multiflora* [7]; this is not sufficient for the representation of this large and diverse group of plants. The availability of the mitochondrial genome of buckwheat is also important for practical applications. The cytoplasmic male sterility (CMS) is associated with rearrangements in the mitochondrial genome. They create chimeric open reading frames (ORF) whose products interfere with the functioning of the mitochondrial electron transfer chain and are thus toxic for cells (for a review, see [8]). The characterization of mitogenomes of CMS lines allowed the finding of events presumably responsible for the sterility in many agricultural plants (e.g., [9,10]). CMS was reported in buckwheat [11]; however, its genetic basis is unknown. Additionally, extranuclear genomes, due to their uniparental inheritance, are an important source of information on the maternal lineage of a species or cultivar. Interspecific hybridization is a promising tool for breeding as it would allow the transfer of beneficial traits (self-compatibility, resistance to abiotic stresses) from related species (*F. homotropicum*, *F. tataricum*) [12]. The characterization of extranuclear genomes of all species involved in breeding will help to trace the genealogy of the hybrids. In this study, we generated the reference genome sequence for the buckwheat mitochondrial genome and studied the diversity of the mitochondrial and plastid genome in 11 buckwheat cultivars (nine Russian, one Japanese, and one Canadian) and the ancestral subspecies, *F. esculentum* ssp. *ancestrale*.

## 2. Results and Discussion

### 2.1. Genome Assembly: Standard and Custom Tools

The initial assembly was performed using Unicycler [13], an assembly tool optimized for the circular genomes. According to Unicycler, there were 10 circular contigs with a similar read coverage that ranged from 733.16 to 1835.25 and a total length of 404 Kb. The BLAST analysis indicated that these 10 contigs are mitochondrial, based on the presence of typical plant mitochondrial genes and on high similarity with the *Fallopia multiflora* mitogenome. These results are unexpected. Despite plenty of evidence that plant mitogenomes can exist in the form of multiple circles and even non-circular forms due to intramolecular recombination mediated by the repeats [14], the single circular molecule, which includes all the subcircles (so-called master circle), is usually recovered in the genome assemblies. This is true especially for the cases where the data are not limited by the shotgun short-read libraries but include mate pair libraries and/or long read data that allow resolving of long repeats. There are several reports of bipartite mitogenomes [15,16] including in Polygonaceae [7]. The higher number of circles is much rarer, in particular a number of circles more than 10 was found only in 4 out of 307 assembled plant mitochondrial genomes deposited in NCBI GenBank (as of 21 February of 2020). All of them represent very special cases: Extremely enlarged mitogenomes in the genus *Silene* [17] and parasitic plants *Lophophytum mirabile* [1] and *Cynomorium* species [18], in which a large part of the mitogenome is acquired by HGT from their hosts. Therefore, initially, we supposed that this result could be a misassembly. To check this, we developed a new assembly tool; it is called Elloreas (ELongating LOng REad ASsembler). It is based on principles similar to NOVOPlasty [19], a seed-and-extend algorithm optimized for the assembly of plastid and mitochondrial genomes out of whole genome sequencing data. While NOVOPlasty was created for short-read assembly, Elloreas performs best with long reads, though it can work with short reads too. An important feature of Elloreas is that it indicates the presence of alternative paths of the extension (in case if they exist).

Basically, Elloreas works in the following way:A user provides a starting sequence, which may be a contig from another assembly or just a random sequencing read.Elloreas maps reads to the 3′ end of this sequence.It finds reads that overlap the ends, such that a part of a read is mapped to the 3′ end while the other part overhangs the contig.Elloreas calculates sequence consensus for these “overhanging parts”.It extends the contig using this consensus.It repeats all steps from “2.” to “6.” for this extended contig.

The work of Elloreas is regulated by multiple parameters, which can optionally be changed by the user, for example, the minimum percent identity required to map a read on a contig and the minimum number of reads supporting an extension required to extend the contig. We used contigs (identified as mitochondrial based on the BLAST search) assembled by Canu and Falcon, two widely used long-read assemblers, as starting sequences. The extension of these contigs by Elloreas showed that they correspond to circular chromosomes: After several iterations of extension, the parts on the 5′ and 3′ ends were found to be identical. Additionally, Elloreas indicated that during the extension there were no “forks”, i.e., several alternative extensions supported by similar amounts of reads. This confirmed the existence of distinct circular chromosomes inferred by Unicycler. The sequences of Elloreas and Unicycler contigs were identical with one exception, an 857-bp deletion in Unicycler contig mito2. The mapping of the raw reads on Unicycler contigs showed that the variant assembled by Elloreas is the correct one. We expect that Elloreas will be useful for the assembly and assembly tests of other small genomes, whether circular or not.

### 2.2. Genome Structure and Gene Content

The result of the assembly was the set of 10 contigs with total length 404,063 bp. Their coverage ranges from 713 to 988× (Table 1). The coverage along all 10 contigs is rather uniform (Appendix A).

Moving 10 kbp from the end of each contig to its start and mapping reads to such forms of contigs also indicates a uniform coverage, implying that these contigs correspond to circular sequences. Long reads generated by the Pacific Bioscience and Oxford Nanopore Technologies platforms are a great tool for the detection of alternative structural variants in mitochondrial genomes (see, e.g., [20]). We did not identify the alternative variants at high frequency; only two variants were supported by >10% reads. The most frequent is an inversion within the chromosome mito9; it is supported by approximately 16% of reads. All other variants were found at a much lower level (see Appendix A). The predominant type of the structural variation is the chromosome merge (25 out of 42 variants with frequency higher than 1%). Buckwheat mitochondrial contigs have a number of repeats; in particular the largest contig, mito1, carries a large direct repeat (~10 Kb). There is also a number of smaller repeats, both direct and inverted (see Figure 1). The repeats are known to be a hotspot for the recombination (see, for example, [21,22]). Indeed, we found that many structural variants, predominantly chromosome merges, are associated with the repeats (Appendix A). This shows that the buckwheat mitochondrial genome undergoes recombination, which generates a diversity of subgenomic forms.

However, these alternative forms have a lower frequency (there are no structural variants supported by more than 50% of reads). This suggests that 10 independent circular chromosomes are the predominant form of the mitochondrial genome of *F. esculentum*. The chromosomes mito1–10 carry a complete set of genes typical for plant mitogenomes: Nine *nad* genes, two *sdh* genes, *cob*, *cox1*–*3*, five *atp* genes, four cytochrome c maturation factors, *matR*, *mttB*, and ribosomal protein (RP) genes (Figure 1).

The latter are variable within angiosperms; virtually any of the RP genes are lost in one or more angiosperm lineages [23]. In particular, the loss of *rps13* occurred in the common ancestor of rosids and *rps8*, in the common ancestor of seed plants. Congruent with this, in *F. esculentum*, the typical mitochondrial *rps8* is absent while *rps13* is present, as well as *rps1*, *3*, *4*, *7*, *10*, *12*, and *14*. Concerning large subunit RP genes, *rpl5* and *rpl16* are present while *rpl2* and *rpl10* are absent. *rpl2* is present in *Nepenthes* [24] but absent in *Fallopia* [7] and *Beta* [25], indicating multiple losses in Caryophyllales. *rpl10* is a pseudogene in *Nepenthes* and completely absent in *Beta* and *Fallopia*. It is not uncommon for mitochondrial RP genes to be replaced by their plastid counterparts, either nucleus-encoded [26] or integrated in the mitochondrial genome [27]. The *F. esculentum* mitogenome carries large insertions of DNA of plastid origin (MIPT) (see Appendix A). They are especially abundant in chromosome 7 where they constitute as much as 42% of its the total length. We assume that the MIPT are recent, based on their high level of similarity with the plastome (94–100%) and on the fact that they are not shared with other Caryophyllales. The MIPTs include a number of plastid genes with intact ORFs (e.g., *rpl16*, *infA*) while many others have internal stop codons (*rps3*, *rpl22*, *rpl14*, *rps8*), making them unlikely to have functional significance. Not only the gene content but the intron content is also variable in mitochondrial genes. There are several mechanisms responsible for this, in particular, retroprocessing—the integration in the genome of the cDNA corresponding to the processed transcripts [28] and horizontal DNA transfer [29]. In *F. esculentum*, *cox1* intron, which was frequently acquired via HGT in many groups of angiosperms [30], is absent, as well as in other Caryophyllales. Five out of nine *nad* genes have introns, and three (*nad1*, *nad5*, *nad2*) of them have interchromosomal trans-splicing. In *nad1*, *atp6*, and *nad4L*, we found an atypical start codon ACG, suggesting RNA editing. The ACG start codon in these genes is also observed in most angiosperms, including Caryophyllales; editing converting it to a typical start codon was shown experimentally in several species [31,32]. Regarding RNA-coding genes, we found 3 rRNA genes and 24 tRNA genes. The latter divide into two groups, typical mitochondrial and chloroplast-like genes. There are 15 native mitochondrial tRNA genes (12 are unique) and 9 chloroplast-like genes (8 are unique). The genes coding for trna-Met (elongator) are duplicated.

### 2.3. Phylogenetic Analysis

Mitochondrial genes, due to their low substitution rate, are a valuable source of information for phylogenetic analysis at a high taxonomic level (see, e.g., [33]). Phylogenetic analysis based on a set of core mitochondrial genes in 28 species representing all major clades of angiosperms and an outgroup resulted in a tree with a topology congruent with ones based on nuclear and plastid data. In particular, Caryophyllales are resolved as a sister group to asterids and *Fagopyrum* is a sister to *Fallopia* (Figure 2).

While it is natural to expect the identical (or highly similar) topology of phylogenetic trees based on plastid and mitochondrial genes, this cannot be taken for granted (see, e.g., [34]). Mitochondrial genes are often acquired by horizontal gene transfer from other plant species. The most notable example is *Amborella trichopoda* [2] and parasitic plants, but the transfers of single genes and gene parts is known in many other species [35,36]. The phylogenetic analysis of single genes helps to reveal the cases of HGT via the incongruence of gene trees and species trees. In order to check for the presence of HGT, we performed phylogenetic analysis of single genes. In all of them, *Fagopyrum* was either grouped with *Fallopia* or unresolved (Appendix A); both cases are consistent with vertical transmission, thus we conclude that no buckwheat genes were subjects of HGT.

### 2.4. Genetic Diversity of the Mitochondrial and Plastid Genome in Common Buckwheat

The assembled sequence of the buckwheat mitogenome allowed us to estimate the diversity among cultivars. We found that it is unexpectedly low (Appendix A). Only three-four substitutions differ Dasha from most other Russian cultivars. The most divergent are the cultivars Koto and Shinanonatsusoba, which have six substitutions. All variants are located in the non-coding regions. Interestingly, three out of six substitutions in Koto are shared with Russian cultivars and two more are shared with Shinanonatsusoba, the Japanese cultivar. This may indicate that Koto and Shinanonatsusoba have a recent common maternal ancestor. Shinanonatsusoba is a Japanese cultivar that originated in 1984 (Nishimaki Et al., 1984). Koto is a cultivar produced by the Canadian breeding company Kade Research in 2002. According to Ikeda [37], Canadian buckwheat breeders (not excepting Kade Research) specialized in the breeding of buckwheat for Japanese market, even the names of many Canadian cultivars (including Koto) are derived from Japanese. Thus, it is highly likely that Japanese cultivars, in particular Shinanonatsusoba, might contribute to the Koto genome. In contrast to the low divergence between cultivars, *F. esculentum* ssp. *ancestrale* differs from Dasha in 220 positions. The prevalent type of the variant is single nucleotide variation (SNV), followed by deletions and insertions (Appendix A). All variants are located in the non-coding regions or in the synonymous positions of the codon and do not affect the amino acid sequence.

In an earlier study [5], we characterized the plastid genome of *F. esculentum* ssp. *ancestrale*. However, we found that it contains a number of inaccuracies, mostly indels associated with the homopolymer and low-complexity regions, which are error-prone for Sanger sequencing. Since indels in the reference genome may result in mapping and variant calling errors, we decided not to use this sequence as a reference. Taking advantage of the long reads, we assembled and annotated the plastid genome of the Dasha cultivar and used it as a backbone for the analysis of the plastid genome diversity. The diversity among cultivars is also extremely low; as well as for the mitochondrial genome, this concerns not only Russian cultivars but also Koto and Shinanonatsusoba. For Dasha and Dizajn, we found no differences at all. Dasha and other cultivars differ in one position: T to G substitution in the intron of the trnG-UCC gene. Shinanonatsusoba carries an additional change, the substitution in the trnG-trnM spacer. As well as for the mitochondrial genome, the number of SNPs with *F. esculentum* ssp. *ancestrale* is much higher: 159 (Appendix A) (148 if SNPs located in the IR counted once). The SNP density is the highest in the small single copy region and the lowest in the inverted repeat (IR), as expected. In total, 81 SNPs are located in the spacers, 20 in the introns, and 58 in the coding regions. What is surprising is that 32 substitutions are non-synonymous (Table 2). While several types of mutations (C-to-T substitutions) can potentially be silenced by RNA editing (see, for example, [38]), only three substitutions out of 32 are C-to-T. Most substitutions affect highly variable genes, such as *matK*, *ycf1*, and *rpoC2*; however, there are also ones that change the amino acid sequence of highly conserved genes encoding photosystem components (*psbA*, *psbH*, *psbC*). While missenses in photosynthesis-related genes usually adversely affect photosynthesis, there are several examples of a positive effect compared to the wild type [39]. We hypothesize that this mutation(s) increases the photosynthesis efficiency and their fixation in buckwheat cultivars is the result of artificial selection. The increase of the photosynthesis efficiency has indeed been an important trend in buckwheat breeding since the middle of the 20th century. The survey of photosynthesis efficiency in different cultivars and landraces showed that in modern cultivars (developed in 1990–2010, including three sampled in our study: Demetra, Dizajn, and Devyatka), it is higher than in old cultivars and local landraces up to approximately 20% [40]. At the same time, modern cultivars have a low adaptive potential, and they are less resistant to unfavorable environmental conditions (in particular, drought) [41].

Taken together, the analysis of the extranuclear genome diversity shows that buckwheat cultivars sampled in this study share a very recent common maternal ancestor. This would not be surprising for Russian cultivars, but our study also included Japanese and Canadian (presumably of Japanese origin) cultivars. This might indicate the loss of genetic diversity as a side effect of the intensification of buckwheat breeding in the last several decades. A similar situation is observed for sunflower: While a high diversity exists in wild populations, most cultivated varieties arise from a limited pool of germplasm [42,43]. This calls for better characterization of the buckwheat germplasm in order to understand the patterns and limits of its diversity (including that of extranuclear genomes). We expect that the availability of reference sequences for plastid and mitochondrial genomes will facilitate this research program.

## 3. Materials and Methods

### 3.1. Data Source, DNA Extraction, and Sequencing

For 10 buckwheat cultivars (Dasha, Dizajn, Demetra, Devyatka, Dialog, Bashkirskaya krasnostebelnaya, Karadag, Kujbyshevskaya, Kazanka Russian cultivars and Shinanonatsusoba Japanese cultivar) and *F. esculentum* ssp. *ancestrale*, the data were generated in this study. DNA was extracted using the cetyl trimethylammonium bromide (CTAB)-based method [44] from a single plant (Dasha, *F. esculentum* ssp. *ancestrale*) or a pool of five plants (other cultivars). For Dasha cultivar, sequencing was performed using a Sequel II instrument (Pacific Bioscience, Menlo Park, California, USA) at DNALink company. For other cultivars (except for Koto) and *F. esculentum* ssp. *ancestrale*, libraries were prepared using a TruSeq DNA library prep kit (Illumina, San Diego, California, USA) and sequenced on a HiSeq2000 instrument. The reads are deposited in the NCBI database under bioproject number PRJNA627307. Koto sequence data were taken from the study [4], DRA accession number DRR046985.

### 3.2. Read Preprocessing for De Novo Assembly

Before assembly, we performed downsampling (the reduction of the number of reads). This was done in order to reduce the amount of CPU and RAM resources required for the assembly. For PacBio data used for the de novo assembly, we found based on the k-mer frequency that the whole set of reads results in a coverage of approximately 800× for the mitochondrial genome. Then, we performed two stages of the downsampling. First, using SeqTk v.1.3. (https://github.com/lh3/seqtk), we randomly picked (command “seqtk 0.1”) 253,638 reads. Second, with the command “kmer_filter” from the modified version (see below) of Stacks 2.5 [45], we removed all reads with a median copy number of k-mers less than 5 (this removes the majority of reads corresponding to single-copy nuclear regions). This operation reduced the number of reads approximately twofold—from 253,638 to 133,619. By default, Stacks removes reads where 80% of k-mers have a copy number below a given threshold. We slightly changed its source code, so the required percent of low-copy k-mers to discard a read was 50%, which means that our version of Stacks uses a median copy number of k-mers in a read as a criterion.

For Illumina data, we randomly picked 50–200 millions of paired reads. As we found empirically, by sampling and mapping different numbers of reads (5–300 millions), this number of reads is sufficient to provide >1000× coverage for the plastid and >100× for the mitochondrial genome.

### 3.3. Mitochondrial Genome Assembly and Assembly Check

The initial assembly was performed by Unicycler 0.4.8 [13] with the default parameters. After assembly, we estimated the mean read coverages of all contigs by mapping all reads to contigs with minimap 2.17 [46] with the option “asm20”, recommended by the author of minimap for mapping PacBio CCS reads. The list of coverages for all contigs was created based on the mapping, by the command “DepthOfCoverage” from GATK 3.8 [47]. Potential mitochondrial contigs were identified based on a similarity search using BLASTN 2.9.0 [48] (e-value threshold of 10^−3^, word size 7 bp). The following genomes were used as query: The mitochondrial genomes of *Fallopia multiflora*, the only publicly available assembled mitochondrial genome from the family Polygonaceae, *Nepenthes × ventrata* (N. *ventricosa* × *Nepenthes alata*) and *Arabidopsis thaliana* and the plastid genome of *F. esculentum*. The latter is necessary in order to distinguish between plastid and mitochondrial contigs and to find regions of plastid origin in the mitochondrial genome. The check of the assembly was performed using dedicated genome assembler, which we named Elloreas (abbreviation for ELongating LOng REad ASsembler). It is deposited at https://github.com/shelkmike/Elloreas. Elloreas was run with parameters “sequencing_technology hifi_pacbio”, ”minimum_length_of_mapped_read_part 8000”, ”minimum_read_similarity 99%”, ”contig_edge_size_to_use_for_mapping 15000”. As starters for Elloreas, we used several random mitochondrial (based on the similarity search described above) contigs from assemblies created by Canu 2.0 [49] and Falcon 1.1.0 [50]. The assembly of Canu was performed with all reads, the parameter “genomesize” was set to 1.5 Gbp, which is the approximate nuclear genome size of *F. esculentum* [51]. The assembly by Falcon was done using reads with a median k-mer coverage of at least 50. The estimated genome size for Falcon was initially set to 1.5 Gbp, but the sum of produced contigs lengths was only 400 Mbp, due to the removal of many nuclear reads by Stacks. Therefore, we reran the Falcon, setting the estimate of the genome size to 400 Mbp. Other parameters of Falcon were set to the values recommended by the authors of Falcon for CCS reads.

### 3.4. Detection of Structural Variants

Detection of structural variants in the mitochondrial genome was performed using the approach described by [52]. It consists of two main steps: Mapping by NGMLR 0.2.8 and then structural variant detection by Sniffles 1.0.11. NGMLR was run with the option made for typical PacBio reads (“presets pacbio”), as it has no dedicated option for PacBio CCS reads.

### 3.5. Plastid Genome Assembly

To assemble the Dasha plastid genome, we took the largest plastid contig (found based on similarity with *F. esculentum* ssp. *ancestrale* plastome) in the Unicycler assembly described above and extended it by Elloreas, which created a circular sequence after 50 iterations of extension. Elloreas was run with parameters “sequencing_technology hifi_pacbio”, ”minimum_length_of_mapped_read_part 8000”, ”minimum_read_similarity 100%”, “contig_edge_size_to_use_for_mapping 15000”. To find possible misassemblies, we mapped PacBio reads by minimap2 with the “asm20” parameter and performed variant calling by FreeBayes v.1.3.0, searching for variants with a variant quality at least 40 supported by more than 50% reads. We found one misassembly in a poly-A region; that misassembly was introduced by Unicycler. An alternative variant (5 adenines instead of 4) was supported by 98.6% reads mapping to this position and we fixed the genome accordingly.

A pairwise alignment by the online version of NCBI BLAST showed that this assembly was collinear with the plastid genome of *F. esculentum* spp. *ancestrale*. The plastome sequence is deposited in NCBI Genbank under accession number MT364821.

### 3.6. Annotation

For the mitochondrial genome, the annotation of protein-coding genes and rRNA genes was performed based on the BLAST search of genes encoded in the mitochondrial genomes of other Caryophyllales (*Beta vulgaris*, *Nepenthes × ventrata*, *Fallopia multiflora*). For the plastid genome, the annotation was transferred from the sequence of *F. esculentum* ssp. *ancestrale* plastome (NCBI accession NC_010776) using GATU [53] with manual correction. Annotation of tRNAs was performed using the tRNA-scan-SE server with sequence source = other mitochondrial for mitogenome and mixed for plastid genome and search mode = default.

The maps of mitochondrial and plastid chromosomes were drawn by a custom script, which utilized Circos 0.69 [54]. Repeats in the mitochondrial genome were found by BLASTN 2.9.0 without a requirement for the maximum e-value but with requirements for the minimum length (500 bp) and percent identity (95%). MIPTs were found by the same method with an identity cut-off of 90% and length cut-off of 100 bp. Since mitochondrial genomes are known to have tRNAs of plastid origin and genes with homologs in the plastid genome (*atp*, *rrn*), only hits that do not overlap with annotated mitochondrial protein-coding or RNA genes were reported.

### 3.7. Alignment and Phylogenetic Analysis

Sets of individual genes were aligned using MUSCLE [55] (version 3.8.31) with default parameters. For the analysis of the combined dataset, alignments of individual genes were concatenated using custom script. Phylogenetic analysis was performed using RaXML software (version 8.2.4) with parameters “-m GTRCAT -× 123, 456 -N 100 -p 098765”.

### 3.8. Mapping and SNP Calling

As a reference for mitochondrial and plastid genome analysis, we used mitochondrial and plastid contigs of the cultivar Dasha generated in this study. Before mapping, the Illumina reads were downsampled to 50–200 million of paired-and reads, in order to minimize the representation of the numts and nupts. The mapping was performed using CLC Genomics Workbench v.9.5.4 with the following settings: match score 1, mismatch cost 3, linear gap cost, insertion cost 3, deletion cost 3, length fraction required to be aligned 1, similarity fraction 0,97, non-specific match handling—map randomly.

SNP calling was performed using the same program with the following options: ploidy 1, ignore positions with coverage above 10,000 (this mean actually no upper limit because in all mappings the highest coverage was lower than 10,000×), ignore broken pairs, minimum coverage 20, minimum count 15, minimum frequency (%) 75, base quality filter = yes, neighborhood radius 5, minimum central quality 20, minimum neighborhood quality 15, read direction filter = yes, direction frequency (%) 30, relative read direction filter = yes, significance (%) = 1, read position filter = yes, significance (%) = 1.

The buckwheat mitochondrial genome contains large inserts of plastid origin. Due to the 10-fold higher coverage of the plastid genome compared to the mitogenome, the mapping of plastid reads on these regions will interfere with the variant calling. In order to get rid of the false variants originated from the mapping of plastid reads, we excluded all variants found in the regions of plastid origin (as defined in Appendix A).

## Figures and Tables

**Figure 1 plants-09-00618-f001:**
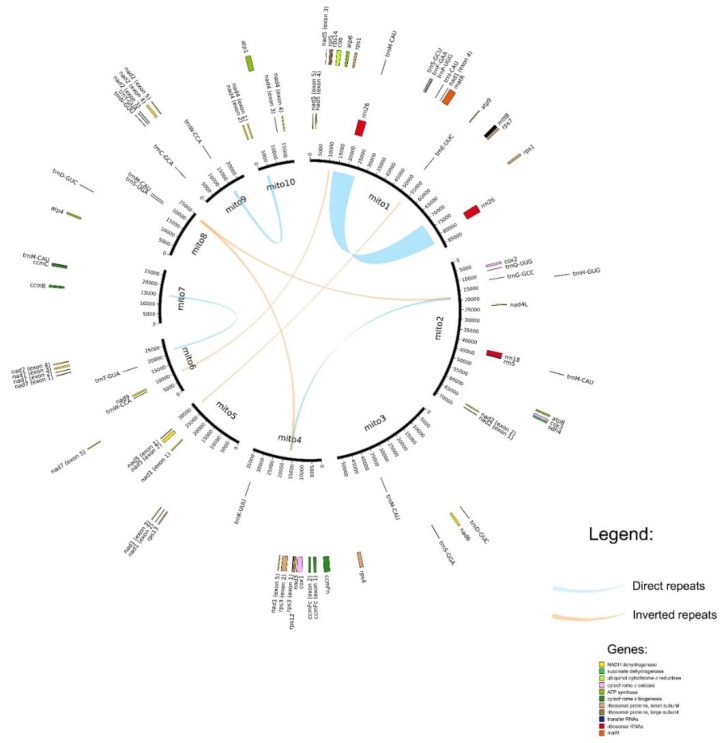
Map of the buckwheat mitochondrial genome representing genes and repeats. Genes shown in the outer circle are transcribed clockwise; in the inner circle, counterclockwise. Repeats with length >500 bp and similarity >95% are represented.

**Figure 2 plants-09-00618-f002:**
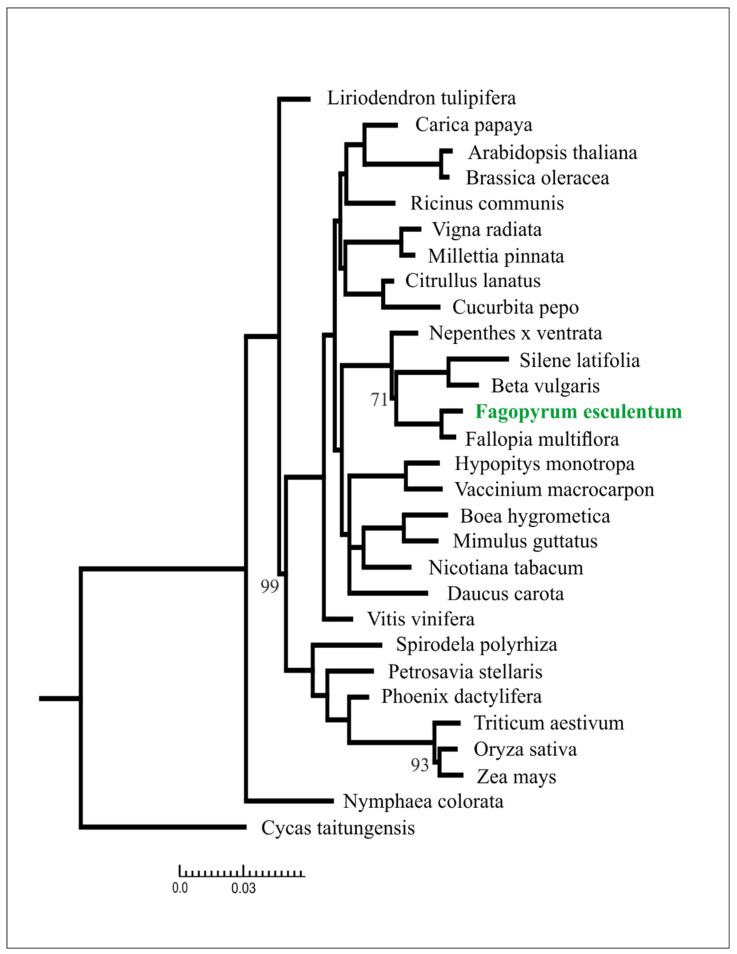
Phylogenetic tree based on maximum likelihood analysis of a concatenated data set of mitochondrial gene sequences. Branch lengths are proportional to the number of substitutions; bootstrap support values are equal to 100, unless otherwise specified.

**Table 1 plants-09-00618-t001:** Length and coverage of buckwheat mitochondrial chromosomes.

Number of Chromosomes	Length	Coverage	Accession Number (NCBI)
1	87,722	861	MT318702
2	71,837	839	MT318703
3	52,654	872	MT318705
4	35,904	763	MT318701
5	31,499	910	MT318704
6	28,895	864	MT318706
7	27,675	713	MT318708
8	25,181	769	MT318709
9	23,738	988	MT318710
10	18,958	727	MT318707

**Table 2 plants-09-00618-t002:** Non-synonymous substitutions that differentiate Dasha and *F. esculentum* ssp. *ancestrale* plastid genes (substitutions located in the IR are counted once).

Reference Position	Type	Reference (Dasha)	Alternative (*F. esculentum* ssp. *ancestrale*)	Coverage	Frequency	Amino Acid Change
1354	SNV	G	T	164	100	psbA:p.Leu49Ile
2105	SNV	A	C	144	100	matK:p.Phe465Val
2809	SNV	C	T	163	99,4	matK:p.Arg230Gln
3467	SNV	C	T	155	98,7	matK:p.Asp11Asn
11370	SNV	C	A	143	100	atpF:p.Lys157Asn
19716	SNV	A	T	184	100	rpoC2:p.Phe85Ile
34532	SNV	C	A	178	100	psbC:p.Leu209Ile
46072	SNV	G	T	161	100	rps4:p.Ser148Tyr
58194	SNV	T	G	188	98,9	accD:p.Phe61Leu
58536	SNV	A	C	149	98,7	accD:p.Glu175Asp
69342	SNV	G	T	129	100	rpl20:p.Ser114Tyr
75628	SNV	T	G	203	100	psbH:p.Ser48Ala
81203	SNV	C	T	152	100	rps8:p.Ser78Asn
83622	SNV	A	G	173	100	rps3:p.Met199Thr
90367	SNV	C	A	146	93,8	ycf2:p.Gln974Lys
92722	SNV	A	G	215	83,7	ycf2:p.Ser1759Gly
110512	SNV	T	G	180	99,4	ycf1:p.Phe278Leu
111003	SNV	C	A	139	100	ycf1:p.Ser442Tyr
112085	SNV	T	C	138	100	ycf1:p.Phe803Leu
113218	SNV	T	A	131	97,7	ycf1:p.Phe1180Leu
115685	SNV	G	A	144	93,1	ndhF:p.Leu687Phe
115832	SNV	A	G	118	96,6	ndhF:p.Phe638Leu
116149	SNV	G	A	142	96,5	ndhF:p.Ala532Val
116840	SNV	C	G	214	98,6	ndhF:p.Val302Leu
118832	SNV	C	G	79	100	rpl32:p.Arg49Gly
127083	SNV	G	T	225	100	ndhA:p.Ser92Arg

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
