# Peer review of "Mitochondrial Genome of *Fagopyrum esculentum* and the Genetic Diversity of Extranuclear Genomes in Buckwheat"

_plants, 2020, doi:10.3390/plants9050618_

Round 1
Reviewer 1 Report
This well-written manuscript provides new genomic sequences for the buckwheat mitochondrial and plastid genomes which the authors then utilize to investigate genetic relationships among cultivars and ancestors. The multiple approaches use to document the assembly of the mitochondrial genome sequence gives high confidence of the 10-part circular maps.
The authors might address the following minor revisions:
Lines 212-215
The authors could mention that:
Among the non-synonymous substitutions only 3 are C-to-T and might possibly become synonymous by RNA editing
Of the three non-synonymous substitutions in the photosynthetic genes, 2 result in conservative amino acid substitutions and are less likely to affect function or process The third (serine to alanine in psbH) might have functional consequence if the serine constituted a phosphorylation site. Is there any information about serine in this context?
Editorial:
Line 66: correct spelling to “similarity”
Line140: italicize rpl10
Line 153: italicize nad1, nad5, nad2
Table 1: The authors should correct the number on the Table itself (Table 2 is labeled Table 1 in the heading) and expand the right column width
Reviewer 2 Report
The reviewed manuscript presents results of a sequencing study of Fagopyrum esculentum. Dasha cultivar was sequenced using long read PacBio Sequel II sequencing, while 9 other cultivars and a WT were sequenced on an Illumina HiSeq2000 machine. Focus was on mitochondrial genome assembly, where the authors put a lot of effort into producing the best possible assembly, especially to determine the basic genome structure (linear v. circular; how many separate structures). This included writing their own targeted genome assembler Elloreas.
I found the experiment to be well designed and carried out. The writing is also very clean with good flow of ideas from one topic to another. Unfortunately a few key pieces are missing for me to believe, or rather be able to verify the main conclusion of the manuscript - that the mitochondrial genome is indeed multipartite, made up of 10 separate circular DNA molecules, as opposed to some alternative arrangement. The missing pieces include:
1) A better critical description of the Elloreas software. For example, one can imagine that its ability to output alternative elongation paths would depend on the number, similarity and length of repeats present in the assembled reads, the average read length, length distribution and the chosen parametrization of the assembly algorithm. It would be much easier to accept the conclusions if we knew the software works well on some kind of simulated data with varying presence of circles, for example. Not just from reading how it works but from actual numerical simulations and experiments.
Also, did some of the alternative assemblies reported by Elloreas (page 4 line 124 on) possibly support a lower number of circles, or were they other kinds of alternatives?
Since this is apparently the first time this software is used for practical purposes, I suggest an entire subsection is added to Methods or some other appropriate section of the manuscript to address these doubts.
2) The identity/sequence of the contigs that were used to grow the Elloreas assemblies. While the authors state that no alternative elongations were produced by Elloreas itself, it is critical to also know, that the initial contigs were disjunctive, not influencing somehow the main conclusion of the study. Also, we do not no what the threshold for these "no alternative elongations" was.
3) The produced assemblies - I could not find them in NCBI yet under the MTXXXXXX accession numbers (I wanted to check points 4 and 5 myself).
4) Were any repeats found? How many, how long, what's their placement in the genome? This is important, since they are the potential regions of recombination and could have decisive role in the overall genome arrangement (or alternative arrangements).
5) Are there any similarities in genome organization to known mitochondrial genomes in plants, possible some that are related to Fagopyrum? Can the locations of potential circle fusion/separation be identified by similarity in gene order or presence/absence of certain genes and repeats?
I am also not sure if the authors considered all previously published data on this topic, e.g. a paper by Zhang et al., 2017 in Molecular Plant 10:1224-1237 that states:
"The mitochondrial DNA of Pinku1 was partially assembled into a contig of 489 827 bp (Supplemental Methods). The base accuracy of theassembled Pinku1 sequences from the SMRT reads was estimated to be 99.96% using Illumina short reads, with even fewer errors in the genic regions (Supplemental Tables 6 and 7)."
Also, Wang et al., 2014 in PLoS ONE 9(8): e105748 show how mitochondrial genomes can be shown/segmented as master circles and subgenomes within those, the separation being mediated by the presence of repeats that may hinder assembly or even cause physical partition of the genome into several circular parts, suggesting that mapping any assembled repeats could be important for reaching the right conclusions.
In conclusion, as a major revision I would recommend either stronger support for the main conclusion (exactly 10 circles), or downtoning the conclusion to "the most likely arrangement", perhaps something in line with Gualberto et al., 2014, such as multipartedness caused by "frequently recombining pairs of repeats", if these can be identified from the data. My main worry here being that the 10 circles might be a mathematical result of observing many alternative arrangements with less than 10 circles, recombined in different ways.
Minor issues:
The authors write:
"By now the only species of Polygonaceae with sequenced mitochondrial genome is Fallopia multiflora;" <- Is there a citation for this to use here? Maybe [14] used several lines later?
Fix grammar:
"We did not identified..." <- identify
Reviewer 3 Report
Please check the number of buckwheat cultivarss, the description in materials and methods (line 242) differs from the introduction (line 58).
Interesting, well-written paper, I have no comments except mentined above checking the cultivars.
Author Response
Comments and Suggestions for Authors
Q: Please check the number of buckwheat cultivarss, the description in materials and methods (line 242) differs from the introduction (line 58).
A: It differs because 11 cultivars were included in the analysis but out of them 10 were sequenced in this study and one – Koto - was taken from an earlier study by Yasui et al.
Round 2
Reviewer 2 Report
In response to my previous suggestions the authors explained why they prefered
to keep a detailed analysis of Elloreas out of this manuscript and went with
downtoning the conclusion about the specific number of circles. This is
acceptable, although I think the abstract also needs to be part of this
redressing. I therefore suggest also the following change in the abstract:
p1 l16 "it consists of 10 circular chromosomes with total length 404 Kb."
In line with the suggested downtoning of the conclusions, I suggest
the authors change this claim in the abstract to something like:
"the prevalent detected form consists of 10 circular chromosomes with a total length 404 Kb."
There was a language issue that remained unchanged from the previous version
and a few more I missed previously:
p1 l31 In contract to plastid genomes <- "in contrast"
p2 l65 Despite there is plenty of evidence <- drop "there is"
p3 l112 We did not identified <- "identify"
p3 l123 there is no structural variants <- "there are"
p6 l172 are often acquired by the horizontal gene transfer <- drop "the"
I consider the other changes satisfactory and see the manuscript as fit for publication.
Author Response
Thank you very much for the comments. We changed the statement in the abstract to "Using long reads generated by SMRT-CCS technology we assembled buckwheat mitochondrial genome and detected that its prevalent form consists of 10 circular chromosomes with a total length 404 Kb" and corrected language errors.